# The Methods of Digging for “Gold” within the Salt: Characterization of Halophilic Prokaryotes and Identification of Their Valuable Biological Products Using Sequencing and Genome Mining Tools

**DOI:** 10.3390/genes12111756

**Published:** 2021-11-01

**Authors:** Jakub Lach, Paulina Jęcz, Dominik Strapagiel, Agnieszka Matera-Witkiewicz, Paweł Stączek

**Affiliations:** 1Department of Molecular Microbiology, Faculty of Biology and Environmental Protection, University of Lodz, 93-338 Lodz, Poland; paulina.jecz@biol.uni.lodz.pl (P.J.); pawel.staczek@biol.uni.lodz.pl (P.S.); 2Biobank Lab, Department of Molecular Biophysics, Faculty of Environmental Protection, University of Lodz, 93-338 Lodz, Poland; dominik.strapagiel@biol.uni.lodz.pl; 3Screening Laboratory of Biological Activity Tests and Collection of Biological Material, Faculty of Pharmacy, Wroclaw Medical University, 50-368 Wroclaw, Poland; agnieszka.matera-witkiewicz@umed.wroc.pl

**Keywords:** halophiles, biomolecules, metagenomics, bioinformatics, genome mining, biodiversity, hypersaline environments

## Abstract

Halophiles, the salt-loving organisms, have been investigated for at least a hundred years. They are found in all three domains of life, namely Archaea, Bacteria, and Eukarya, and occur in saline and hypersaline environments worldwide. They are already a valuable source of various biomolecules for biotechnological, pharmaceutical, cosmetological and industrial applications. In the present era of multidrug-resistant bacteria, cancer expansion, and extreme environmental pollution, the demand for new, effective compounds is higher and more urgent than ever before. Thus, the unique metabolism of halophilic microorganisms, their low nutritional requirements and their ability to adapt to harsh conditions (high salinity, high pressure and UV radiation, low oxygen concentration, hydrophobic conditions, extreme temperatures and pH, toxic compounds and heavy metals) make them promising candidates as a fruitful source of bioactive compounds. The main aim of this review is to highlight the nucleic acid sequencing experimental strategies used in halophile studies in concert with the presentation of recent examples of bioproducts and functions discovered in silico in the halophile’s genomes. We point out methodological gaps and solutions based on in silico methods that are helpful in the identification of valuable bioproducts synthesized by halophiles. We also show the potential of an increasing number of publicly available genomic and metagenomic data for halophilic organisms that can be analysed to identify such new bioproducts and their producers.

## 1. Introduction

Halophiles are a highly miscellaneous class of extremophilic organisms characterised by their requirements for high salinity and comprise entities from all three domains of life, namely Bacteria, Archaea, and Eukarya [1,2]. Owing to their phylogenetic origin and the nourishment acquisition manner, halophilic microorganisms can be grouped as follows: (1) heterotrophic, phototrophic or methanogenic archaea; (2) heterotrophic, lithotrophic or photosynthetic bacteria, and (3) heterotrophic or photosynthetic eukaryotes [2,3,4]. In the following parts of this article, we will focus only on the halophiles representing the first two mentioned groups.

Due to the salt concentration requirements (specifically and commonly, sodium cations and chloride anions), halophiles can be classified as slight, with optimal growth at 0.2–0.85M (1–5%) NaCl, moderate thriving in 0.85–3.4M (5–20%) NaCl, and extreme growing optimally at 3.4–5.1M (20–30%) [5]. On the contrary, non-halophiles do not grow in the environment containing above 0.2M (1%) NaCl, and halotolerant organisms are viable in the presence or absence of highly saline conditions, but it is not necessary for their optimal growth [5]. Moreover, halophilic and halotolerant organisms are able to adapt to a broad range of salt concentrations, occurring seasonally, annually, or irregularly in their natural environments [4,6].

Halophilic microorganisms are forced to efficiently prevent osmosis due to the high external salinity, and thus they have evolved two types of strategies to struggle with cellular water loss—“salt-out” (“low-salt-in”) and “salt-in” [4,7]. The first strategy is based on biosynthesis (*de novo* or from the storage substances) or absorption from the environment compatible solutes (osmolytes or osmoprotectants) and is utilized mainly by moderate halophiles, halotolerant bacteria, and eukaryotes. Polyols, sugars, amino acids, betaines, ectoines, N-acetylated diamino acids, and N-derivatized carboxamides of glutamine are commonly used. The second strategy relies on the accumulation of salt, predominantly potassium chloride, to provide intracellular osmotic pressure comparable to the external one and is typical for extremely halophilic Archaea and a few representatives of Bacteria (genus *Salinibacter* and members of the order *Halanaerobiales*) [2]. This mechanism requires specific adaptation of enzymes and other proteins, e.g., by elevating a level of negatively charged amino acids, leading to the formation of an acidic proteome as observed in *Halobacterium* sp. NRC-1 [7,8,9]. However, the evidence provided by Elevi Bardavid and Oren (2012) suggests that this may not be a strict rule, and other mechanisms must be involved in osmoregulation in halophiles [10]. Some halophiles (especially from the archaeal class of *Halobacteria*) have applied a mix of these strategies to cope mainly with periodic fluctuation of salinity [11,12].

## 2. Global Distribution of Hypersaline Environments

Although the oceans and seas (average salinity–0.6 M, 3.5% or 35 parts per thousand) come to mind first, the term “hypersaline environments” refers to conditions where the salt concentration exceeds that present in marine basins (even ten times up to or above salt saturation) [13,14]. Hypersaline environments are generally classified as thalassic (thalassohaline) when originating from seawater with its characteristic ionic composition (dominated by Cl^−^—49% and Na^+^—42% of the total molarity) and as athalassic (athalassohaline, also inland or epicontinental) not directly associated with a marine source and dominated by divalent ions mainly Mg^2+^ and Ca^2+^ [5]. Some authors also distinguish the third type—artificial reservoirs employed for salt production (saltern crystalliser ponds) [15]. 

Nevertheless, hypersaline ecosystems and their habitats are widely explored mainly due to their utilisation in mineral processing—salt mines, solar salterns and salt flat [16,17,18], aquaculture (e.g., brine-shrimps predominantly in the Great Salt Lake, commercial lakes in China, Russia, and Kazakhstan) [19,20], biotechnical applications (biomolecules like enzymes, pigments, antimicrobial agents, nanoparticles) [21,22,23,24]; in the role of microbial cell factors [25]; environmental and protection studies as niches for eukaryotes, prokaryotes, and archaea [15,17,26,27], biodegradation of contaminants [28,29,30,31]; astrobiological signification and early Earth connotations [32,33]. On the other hand, issues related to the anthropogenic impact on hypersaline environments have become more and more significant in recent years. To name only some: climate alteration, overexploitation of mining and mineral extraction, overflow of agriculture, water diversion and salinity enlargement, urban overdevelopment, industrial sewage and contamination with ultimate examples of the Dead Sea, the Caspian Sea, the Aral Sea, and the Great Salt Lake [15]. And as it turns out, these activities have a tremendous influence on the (bio)diversity of the hypersaline ecosystems.

## 3. Biodiversity of Hypersaline Environments

The Dutch microbiologist and botanist investigating various saline and hypersaline lakes worldwide Lourens G. M. Baas Becking (1895-1963) claimed that “*everything is everywhere:* but, *the environment selects*” [34]. This statement is highly relevant to the hypersaline ecosystems broadly distributed around the world, from the Antarctic to the Himalayas, from Australia to the USA, from Africa to South America, and thus are much dissimilar in terms of salt concentration, chemical composition, and presence of additional stress conditions designated by geological attributes [5,35,36,37]. Therefore, they are not only characterised by high-salt content but other environmental physicochemical extrema like high pressure and UV radiation, low oxygen concentration, hydrophobic conditions, extreme temperatures and pH, high concentrations of toxic compounds and heavy metals [27,36,38,39,40,41,42].

The most frequently identified bacterial phyla in saline and hypersaline environments are Actinobacteria, Bacteroidetes, Cyanobacteria, Proteobacteria (Alpha, Beta, Gamma, and Delta), and Firmicutes [36,41,43,44,45,46,47,48]. Halophilic Archaea are typically represented by Halobacteria and methylotrophic methanogens class members, both belonging to the phylum Euryarchaeota. The former includes about 70 genera and 260 species, classified in three orders and six families: the *Halobacteriales* (families *Halobacteriaceae*, *Haloarculaceae*, *Halococcaceae*), the *Haloferacales* (families *Haloferacaceae*, *Halorubraceae*), and the *Natrialbales* (family *Natrialbaceae*), and the latter comprises of 4 classes: *Methanomicrobia*, *Methanobacteria*, *Methanopyri* and *Methanococci* [49,50]. Furthermore, it was demonstrated that Archaea tend to dominate Bacteria as salinity increases, which is illustrated by an excellent example of the two arms of the Great Salt Lake, significantly different in the salt content, and thus in a taxonomy of their inhabitants [36,37,51,52]. Moreover, the composition and structure of halophilic communities in saline and hypersaline ecosystems are considerably influenced by the salinity fluctuation in time or geographical location and may differ between the places of sampling within the same setting [53,54,55,56,57].

Saline soils are other fascinating and valuable from the ecological, economical, and biotechnological points of view examined environments with abundance and high diversity of their inhabitants, taxonomically comparable to aqueous ones (phylum level) [58,59]. In addition, it has been established that salinity, along with pH and electrical conductivity (EC), are the pivotal factors determining the variety and arrangement of halophiles and haloalkaliphiles in saline soils [60,61,62]. Intriguingly, these microorganisms are gaining special attention due to progressing global soil salinisation, and thus their potential applicability as plant symbionts enabling and increasing crop productivity in saline soils [63,64]. It is noteworthy that a successful attempt was currently done to employ halophilic microorganisms as bioindicators of the soil salt contamination caused by extensive de-icing of roads during harsh winters in Baltimore, Maryland, USA. It became possible since halophiles become persistent members of microbial communities as a result of salting roads for their de-icing during winter months [65]. 

In addition to these environmental species, there is a constantly extending group of human or human-related halophiles, both bacterial and archaeal [66,67,68,69]. Brining, i.e., treating, food with dry salt or a salt solution, is one of the oldest methods to preserve and season the eatables in food processing. There are numerous and continual scientific reports on isolating new halophilic microorganisms, the diversity and properties of halophilic Bacteria and Archaea, as well as genomic analyses from commercial salt [70,71,72,73], cheeses [74,75,76,77], table olives [78], kimchi (Asian fermented vegetables) [79,80,81], and shrimp paste [82,83]. Recent years have also brought interest in the halophilic and halotolerant prokaryotes contributing to the human gut microbiota [73,84,85,86]. This attention results in part from observing a hazardous tendency to consume increasing amounts of salt delivered with food and its tremendous consequences on human health, including obesity, hypertension, cardiovascular disorders, and stomach cancers [69].

Finally, halophilic prokaryotes are the established producers of multiple biomolecules and chemical substances, predominantly osmolytes, hydrolytic enzymes, and pigments (e.g., carotenoids) [22,87,88,89,90]. The increasing interest in compounds and proteins of halophilic origin results mainly from the fact that they remain active under harsh conditions like high salinity, extreme temperatures, and ultimate pH [91]. Moreover, halophilic enzymes retain solubility and solvation in low water activity [11], and as has been shown recently, they demonstrate anti-desiccation and antifreeze properties, so desirable in food processing and preservation [88]. Halophiles also produce biodegradable polysaccharides and polymers, potentially replacing environmentally hazardous plastics; glycoproteins are considered promising candidates in nanoparticles synthesis, and gas vesicles are examined in terms of an effective drug delivery system as described thoroughly in a review released by Singh and Singh, 2017 [90]. Despite that, bacterial and particularly archaeal halophiles for decades have been underestimated and unexplored in terms of the ability to produce various bioactive compounds, especially of antimicrobial and anticancer potential. However, due to the rapid development of molecular techniques described in the following part of this review, they turned out to be a promising and rich source of diverse biomolecules of great importance in the ongoing post-antibiotic era that is additionally characterised by the galloping increase of cancer cases [22,91,92,93,94,95]. Due to methodological difficulties, time-consuming and expensive procedures that require frequent optimization and the increasing availability of sequencing, research on halophilic biomolecules are moving more and more towards genomic and metagenomic-based bioinformatics analyses [96,97,98,99,100,101].

## 4. Experiment Strategies Used in Halophiles Research

As pointed out, microorganisms living in saline environments are characterized by high diversity, like other extremophiles. A wide range of methods is used to better understand this diversity, including culture-based experiment strategies and culture-independent approaches for direct testing of environmental samples. Studies involving cultivation under laboratory conditions have identified and tested most of the currently known halophiles to obtain their exact characteristics [16,80,102,103]. Through the use of laboratory cultivation of pure bacterial cultures, new bioagents produced by halophiles can also be discovered [95,104,105]. However, these traditional cultivation-based methods do not always work well for halophiles. In the saline ecological niches, numerous species of microorganisms are encountered that cannot be easily cultivated in laboratory conditions. Often, only less than 1% of microorganisms from a given site can be successfully grown in the laboratory [106]. Thus, new methods of optimal and high-throughput culturing of microorganisms, based on the use of modern approaches such as culturomics [107,108] or optimal culture enrichment [109], are developed. However, there are still many microorganisms that cannot be tested through common-based laboratory analysis. It is also worth noting that the methods based on the culturing of environmental microorganisms usually provide not only complete information about the taxonomic composition of a specific microbiome but also about the proportions in an abundance of individual microorganisms. For precise inquiry regarding the biodiversity of microbiomes inhabiting specific ecological niches, it is necessary to use methods that provide insight into the studied environment without culturing the organisms living in it [110,111].

Culture-independent methods are the primary investigation tools for looking into the microbial “dark matter” based on the sequencing of nucleic acids for both entire microbial communities (metagenomics and metatranscriptomics) and single cells (single-cell sequencing) [110,112]. Furthermore, functional metagenomics can be used, where it is possible to screen environmental samples for the production of substances of interest with potential industrial applications [113,114]. Finally, metatranscriptomic analyses enable the assessment of the gene expression in the studied environment. Thanks to their use, it is possible to identify active metabolic pathways in the environmental niche, which gives a unique insight into its characteristics. The best results are achieved by combining metagenomics with metatranscriptomics analysis, where the former one identifies genes in the studied microbiome and allows the creation of MAGs (metagenome-assembled genomes), while the latter one allows assessing the expression level of the detected genes [115,116]. Despite the great potential of obtaining significantly valuable data in this way, still relatively unpopular experiments due to the great difficulties in receiving the appropriate quality material for research from saline environments due to the low available biomass and a high diversity of the samples are observed [117]. Proteomics also is used in the study of halophilic microorganisms, where gene ontology analysis of some up-regulated proteins determining molecular functions, cellular components or a biological process are identified, but it is usually applied when individual strains are grown under laboratory conditions [118,119,120]. Recently, there have been attempts to use proteomics to directly analyse environmental samples, but this is difficult and still rarely applied. The main reason for the difficulties in using proteomics is the low biomass of proteins in the analyzed samples [115]. In the following part of this article, we will focus on characterising the methods based on nucleic acid sequencing and their role in understanding the biodiversity and biotechnological potential of microorganisms living in saline environments.

## 5. In Silico Methods for Identification of Novel Halophiles Bioproducts

For many years, the availability of methods based on NGS (Next Generation Sequencing) has increased. New sequencing methods such as SMRT (Single Molecule Real-Time) from PacBio or nanopore sequencing from Oxford Nanopore Technologies (ONT) have been also invented and developed. All of them have changed the research insight concerning microbiomes of saline environments as well as the analysis of individual strains thanks to the quality of genomic and transcriptomic data improvement [121,122]. Due to the development of the new and improvement of the already existing sequencing methods that prove obtaining much longer reads is possible, where the maximum length of 300 bp (base pairs) for Illumina, 400 bp for MGI and Ion Torrent, 1200 bp for Roche, and in the case of ONT and PacBio, the length of a single read can reach several hundred thousand bp with a median length up to 100 kbp [71,123,124,125,126]. However, with the ONT technology, the obstacles regarding a relatively low accuracy of reads compared to the second-generation sequencing platforms and even SMRT is observed. In the case of ONT and SMRT, very high requirements are defined in terms of the quantity and quality of the DNA used to prepare the libraries. To avoid the problem of low accuracy, manufacturers are constantly perfecting their technologies both in terms of equipment, reagents and algorithms used in the base-calling process. These actions lead also to a decrease in the cost of ONT sequencing and help to improve the quality of raw data [127,128,129]. Data obtained from sequencing are often used for in silico analyses, where the identification of interesting bioproducts in the form of secondary metabolites, AMP, or enzymes can be performed. For this intended purpose, some bioinformatics methods are used, which are characterised in the following part of this section.

One of the most explored areas in the field of bioproducts prediction from sequencing data is the identification of biosynthetic genes clusters (BGCs). This approach permits the identification of the pathways responsible for the production of secondary metabolites. BGC identification methods are continuously improved. One of the most intensively developed tools in this area is the antiSMASH. The currently released version 6.0 helps to identify 71 types of BGCs including non-ribosomal peptide synthetases (NRPSs), type I and type II polyketide synthases (PKSs), lanthipeptides, lasso peptides, sactipeptides, thiopeptides, bacteriocins, terpenes and much more [130]. For identification of BGCs, the antiSMASH uses the set of “rules” which define the core biosynthetic functions that need to exist in a genomic region to constitute a BGC. Additionally, a rule-independent method based on the ClusterFinder prediction algorithm was implemented to this platform [130]. The antiSMASH is widely used in halophiles research for the identification of BGCs in genomic and metagenomic data with satisfying results. The combination of rule-based and rule-independent methods in a single platform is one of the biggest advantages of antiSMASH in halophiles research. Many BGCs responsible for the synthesis of valuable halophilic bioagents already have been described, which allows the identification of their variants using the rule-based approach with high specificity. However, halophile genomes most likely also contain a large number of unknown BGCs, in the identification of which a rule-independent approach can be very helpful.

As an example of BGCs identification, *Planococcus maritimus* SAMP MCC 3013 may be mentioned, where a BGC for biosurfactant synthesis was identified [131]. In this genome, researchers discovered a complete gene cluster comprising genes *dxs*, *ispC*, *ispD*, *ispE*, *ispF*, *ispG*, *ispH*, responsible for the biosynthesis of terpenes. Based on genomic and functional analysis, the authors coined the term "terpene containing biosurfactant" for the surfactant produced by *Planococcus maritimus* [131]. Analysis of *Planococcus maritimus* SAMP MCC 3013 biosurfactant shows that the compound is active against *Mycobacterium tuberculosis* (IC50 64.11 ± 1.64 μg/mL and MIC 160.8 ± 1.64 μg/mL), *Plasmodium falciparum* (EC50 34 ± 0.26 µM), and cancer cell lines HeLa (IC50 41.41 ± 4.21 μg/mL), MCF-7 (IC50 42.79 ± 6.07 μg/mL) and HCT (IC50 31.233 ± 5.08 μg/mL). This activity suggests its potential as a lead candidate for drug development investigations [132]. Also, other halophilic microorganisms were analyzed similarly and were identified as potential new producers of highly valuable biosurfactants. *Bacillus amyloliquefaciens* was identified as a producer of surfactin and fengycin. Approximately 37.7 kb fengycin biosynthetic gene cluster with 90% similarity to fengycin cluster of DSM 23117 containing *FenA, FenB, FenC, FenD, and FenE* genes was identified in the genome of *Bacillus amyloliquefaciens* [133]. Interesting BGCs were also identified in metagenomes such as the Shark Bay Microbial Mats metagenome where BGC responsible for syringomycin production was identified. In this study, 17 of 28 condensation domains from MAGs belonging to Chloroflexi phylum, aligned closely with syringomycin [134]. Searching for novel halophilic biosurfactants is a promising field of research due to their potential as antimicrobial agents, bio-detoxifiers, and emulsifiers [105,135,136,137,138]. Halophiles producing biosurfactants can also find application in bioremediation, especially in high-salinity conditions [138,139]. Therefore, the identification and in-depth characterisation of halophiles capable of producing biosurfactants are crucial. 

Other important compounds produced by halophiles are ectoine and hydroxyectoine. Compatible solutes such as ectoine and hydroxyectoine are used as protective factors for cells, DNA, and proteins, which makes them highly valuable for cosmetics and medicine sectors [140,141]. Ectoine is commonly used in the cosmetics industry, but clinical trials are also conducted to indicate its importance in the treatment of eye and respiratory diseases [142,143,144,145]. In silico studies for new microorganisms producing ectoine and hydroxyectoine can be improved by using BGC identification methods like antiSMASH [146,147,148]. Microorganisms identified in this way can be an alternative to those currently used. They can also combine the production of ectoine with other bioproducts, such as polyhydroxyalkanoates (PHAs) [147]. As an example of this type of co-production *Salinivibrio proteolyticus* M318 isolated from fermented shrimp paste can be mentioned [147]. The authors identified the first strain of *Salinivibrio proteolyticus*, where the complete phaCAB and teaABCD operons were present in the genome. The phaCAB operon is associated with the production of PHAs. It consists of three genes: polyhydroxyalkanoate synthase (*phaC-orf00667*), acetyl-CoA C-acyltransferases (*phaA-orf00669*), and acetoacetyl-CoA reductase (*phaB-orf00670*), while teaABCD operon is associated with the synthesis of ectoine. It includes the *ectA*, *ectB, ectC,* and *lysC* genes responsible for synthesising ectoine and ectoine hydroxylase gene (*ectD-orf00133*) accountable for producing hydroxyectoine from ectoine [147]. Thanks to the combined production of PHAs with ectoine and hydroxyectoine, *Salinivibrio proteolyticus* M318 is a very valuable strain that can be used broadly in the industrial production of these substances.

BGCs identification can be useful also in the prediction of non-ribosomal produced antimicrobial peptides. The case with streptomonomicin (STM) produced by *Streptomonospora alba* can be proposed. STM is a lasso peptide with antimicrobial properties. It belongs to the class of ribosomally synthesized and post-translationally modified peptides (RiPPs) [149]. STM inhibits the growth of Bacillus anthracis, the causative agent of anthrax, and its MIC was in the range of 4-8 µg/mL. Genomes analysis of bacteria isolated from salt-pans with Plant Growth Promoting Features (microorganisms naturally capable of enhancing plant growth and protecting crops from pests) also determine antimicrobial agent encoding BGCs. Here, identified BGCs were similar to bacteriocin encoding BGCs [150] AntiSMASH was also utilized in the search for other BGCs, such as those responsible for the production of exopolysaccharide and pigment source [151], Persiamycin A [152], or carotenoids [153]. Due to its versatility, antiSMASH performs well in halophilic analysis, but it is usually combined with other rule-based or rule-independent tools for better predictions.

The use of in silico methods for BGCs identification helps in more effective quest of valuable compounds new producers. It is particularly important given the rapid increase in the number of publicly available halophilic genomes and metagenomes. Such data can be screened for the presence of BGCs and then analysed more thoroughly if show significant biotechnological or pharmaceutical potential [154]. It is worth to point out that rule-based methods for identifying BGCs require high-quality input assemblies [155]. When low-quality assemblies or metagenomic data, characterized by low continuity and significant fragmentation, are obtained, tools based on the identification of individual core domains, such as NaPDos or eSNaPD [156,157], are preferable. Also, biosyntheticSPAdes to generate better quality input data for BGCs analysis based on de novo assembly graph analysis can be an alternative solution [158]. The growing number and quality of databases containing information on BGCs also have a positive effect on the prospects for the development of BGC identification methods [159,160,161]. 

Identification of novel valuable enzymes can be also achieved using simple homology-based approaches. In this case, query protein sequences are aligned to enzyme reference databases such as Expasy ENZYME [162], BRENDA [163], KEGG [164] or protein domains and motifs databases like Pfam [165], TIGRFAM [166] or SMART [167]. These approaches can be used for in silico screening and enzymes sequence characterisation identified in laboratory conditions. Many halophilic microorganisms produce hydrolytic enzymes highly tolerant to salinity, thermostable and stable in a broad pH spectrum [168]. Moreover, halophilic enzymes have not only a significant economic value but are also utilized in eco-friendly industrial processes, enhancing their role in sustainable development [24,169,170]. Using simple homology-based comparison can help understand enzymes structure, properties and similarity to enzymes from other taxon’s. As an example, Hypersaline lake “Acıgöl” esterase (hAGEst) and GH11 xylanase Xyn22 can be mentioned [113,114]. Both them were identified and purified based on functional metagenomics methods. Newly discovered hAGEst esterase is similar to the alpha/beta hydrolase from *Halomonas gudaonensis* (WP_089686035.1) with a 91% amino acids sequence identity. It is characterised by high activity at low temperatures, high tolerance to DMSO, and metal ions (majority of analysed metal ions in a concentration of 1 mM improve the activity of the esterase where in the case of other esterases, the presence of these ions cause a reduction in the activity of an enzyme). These features can be important in various industrial applications such as detergent formulations, pharmaceutical production and other fine chemicals [113]. On the other hand, GH11 xylanase Xyn22 has very high halotolerance and thermal stability. GH11 xylanase coding sequence was identified in the metagenomic DNA of a saline-alkaline soil. Based on the amino acid sequence analysis and site-directed mutagenesis, it was shown that acidic amino acid residues E137 and E139 are responsible for halotolerance, while the aromatic interaction between Y48 and F53 is responsible for thermostability. Thanks to their properties, specific enzyme can be used in various fields, like seafood processing, paper industry, or biofuel production, and is a good foundation for further work on modified enzymes of this type [114]. Another example is a novel metagenome-derived halotolerant cellulase PersiCel3 obtained from rumen microbiota can be mentioned [171]. In this case, for new enzyme identification, a multi-stage in silico screening pipeline was employed. Analyses were based on NCBI BLAST alignment of predicted ORFs to sequences from the custom database containing experimentally validated thermostable and/or halotolerant cellulase sequences selected by literature mining. The existence of cellulase domain in predicted genes was confirmed by alignment to sequences from NCBI Conserved Domains Database (CDD) [172]. Maximum activity of the PersiCel3 could be seen in the concentration of 3 M NaCl for both free (132.46%) and immobilized (197.47%) enzymes. Applying both the free and immobilized enzyme during the degradation of the rice straw in saline conditions leads to an increase in the production of reducing sugars [171]. In the field of microbial enzymes identification, more sophisticated, machine-learning-based tools were also developed, but they have not gained much popularity in halophilic research yet [173,174,175]. Most likely, it is related to the simplicity and clarity of the standard analytical pipelines. Another group of in silico methods is associated with ribosomally synthesised antimicrobial peptides’ (AMPs) identification. Few databases and tools dedicated to AMPs identification and prediction can be found. Some general antimicrobial peptides databases like DRAMP [176], CRAMPR [177], APD [178], LAMP [179], and numerous specific for different types of AMPs databases [180] are publicly available for researchers. AMPs identification can usually be performed as a simple sequence alignment process [181], but also more sophisticated methods based on machine learning can be used [182,183,184,185]. The direct identification of AMPs is used rather rarely in halophile genomic and metagenomic sequences. Only a few examples of identification of genes coding AMPs after their wet lab detection have been described [181,186]. Unfortunately, screening of genomic and metagenomic data for AMPs genes identification is rarely performed in the analysis of halophiles. It is a significant gap, especially because the halophiles are well-known producers of AMPs [186,187,188,189,190], and learning more of their products through in silico analysis could lead to the discovery of new effective antibiotics. Performing an extensive screening of AMPs may allow the identification of new valuable bioproducts and halophilic strains that may be their producers, as is the case with other groups of microorganisms [191]. In the context of the rising problem with antibiotic resistance, the identification of novel AMPs to which bacteria do not easily develop resistance may be the last hope [192].

The summary of information about selected bioinformatics tools used to identify new bioproducts in genomes of halophilic microorganisms and saline environment metagenomes has been presented in Table 1.

## 6. Conclusions

In recent years, there has been an intensive development of research methods that allow for better insight into the biodiversity of extreme environments, including high salinity environments. New niches such as brine graduation towers or food products have started to be investigated as profoundly as, so far the most analysed, salt environments, such as salt lakes, brines, or the Dead Sea. The use of modern research techniques based both on the culture-dependent and culture-independent methods enables a better understanding of the potential of halophiles and setting new directions for their applications. The research on halophiles is also important since these microorganisms are often polyextremophiles simultaneously, which is associated with their unique properties and may be significant in the context of astrobiological research, where halophiles are often considered organisms that could be detected on distant planets or moons [199]. 

The intensive development of sequencing methods and related bioinformatics tools opens new doors for identifying compounds of great importance for biotechnology and pharmacy. The falling costs of NGS and the development of third-generation sequencing methods allow us to look at saline environments’ previously inaccessible microbial dark matter. However, much remains to be done in the area of developing new analytical tools. It is necessary to conduct further research on the methods of identifying new compounds that may be of significant importance for biotechnology and pharmacy. An important niche is also the identification of absolutely new enzymes, BGCs, or AMPs that do not show substantial similarity to those that have already been known. It is also worth noting that halophilic prokaryotes are rarely analyzed for the presence of ribosomal AMPs in their genomes. The main focus of genome mining activities is on the identification of BGC. On the other hand, the identification of ribosomal AMPs may be an interesting niche for research due to the fact that there are several examples of AMPs produced by halophiles, and it is quite probable that other compounds of this type will be found in other strains.

It should be noted, however, that despite the need for further development of genome mining methods, so far, thanks to their use, it has been possible to identify many compounds important for humanity, and one can hope that further research will allow for identification of even more of them, which will significantly support green economy and pharmacy. Saline environments and halophiles, thanks to their characteristics, are an important source of new biomolecules identified through the use of the aforementioned methods. Halophiles are known as biotechnologically preferable microorganisms capable of producing heterogenous bioproducts [200]. They have been identified as a relevant source of stable enzymatic proteins, antioxidants, stabilising agents, or antimicrobial compounds such as halocins with additional myocardial protection activity [24]. Research is also being carried out to improve biotechnological applications by utilising genetic manipulations and modifications based on mutagenesis, the introduction of recombinant plasmids, and heterologous gene expression.

In summary, in silico identification of halophile bioproducts is crucial for optimal characterisation of this group of microorganism and their use in the industry. Growing databases containing the genome sequences of the halophilic organisms and metagenomes from saline environments offer the opportunity to perform extensive in silico analyses to identify their so far hidden potential. More efforts should be made to identify new AMPs and their producers among halophiles. This area has been greatly neglected and should be considered more frequently for in silico characterization of the biotechnological potential of halophiles. A positive boost is given by the further development of sequencing technology, leading to improved sequencing data quality. This is particularly important for metagenomic studies, which are a key component in halophile research and facilitate the characterization of the biotechnological potential of uncultivated halophiles.

## Figures and Tables

**Table 1 genes-12-01756-t001:** Strengths and weaknesses of methods used in the in silico identification of bioproducts.

Target	Class of Methods	Bioinformatic Tool Examples	Advantages	Disadvantages
Biosynthetic Gene Clusters	Rule-based BGCs identification	antiSMASH [130]	High number of identified BGC classes (71 in antiSMASH 6.0 version) Manually curated rules User-friendly pipelines (e.g., web interface of antiSMASH)	Requires high-quality assemblies Limited to BGCs for which rules have been implemented
		PRISM [193]		
		SMURF [194]		
		BAGEL [195]		
	Rule-independent BGCs identification	eSNaPD [157]	Can be used for highly fragmented data (e.g., metagenomes) A potential to identify novel BGCs (e.g., EvoMining)	Less specific for known BGCs then rule-based methods
		NaPDos [156]		
		EvoMining [196]		
		ClusterFinder [197]		
Halophilic enzymes	Classic alignment based approach	Expasy Enzyme [162]	The user is not limited by a predefined set of databases and comparison parameters User-friendly web interface and stand-alone versions Very flexible	Requires a combination of tools for the best effect In some cases, a choice of optimal pipeline can be challenging
		Pfam [165]		
		BRENDA [163]		
		KEGG [164]		
		BLAST [198]		
	Automated pipelines	Anastasia [173]	Provides high reproducibility Usually better optimized than classical analytical pipelines Does not require detailed knowledge of all analysis parameters (the default parameters should be appropriate for most common cases)	Limited customization options
		MCIC [174]		
		FINDER [175]		
Ribosomally produced AMP	Classic alignment-based approach	DRAMP [176]	The user is not limited by a predefined set of databases and comparison parameters Very flexible Any combination of databases for analysis can be selected	Requires a combination of tools for the best effect In some cases a choice of optimal pipeline can be challenging Cannot identify the AMP types that are not included in the database
		CAMPR [177]		
		LAMP [179]		
		APD [178]		
	Automated pipelines	Macrel [182]	Provides high reproducibility Usually better optimized than classical analytical pipelines Does not require detailed knowledge of all analysis parameters (the default parameters should be appropriate for most common cases)	Limited databases for individual tools Cannot identify the AMP types that are not included in the database
		AMAP [184]		
		AmPEP [185]		

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
