# Peer review of "The Methods of Digging for “Gold” within the Salt: Characterization of Halophilic Prokaryotes and Identification of Their Valuable Biological Products Using Sequencing and Genome Mining Tools"

_genes, 2021, doi:10.3390/genes12111756_

Round 1

Reviewer 1 Report

The submitted manuscript represents a complete and detailed review about the topic of halophilic microrganisms. However, very little care was attributed to manuscript writing. Text is very long, repetitive and not carefully revised. Some comments (many more could be given as example):

- sections´ numbers are wrong (all starting at 1)
- keywords: the word "keywords" is duplicated
- different types of lettering is used. Example: line 34, line 65, etc.
- Figure legend is not correctly placed

The major critic is that many parts of the text are repetitive. It most certainly result from copy/past of text written by different persons, without a careful revision at the end to combine them.

Reviewer 2 Report

Review: Digging for “gold” within the salt: characterization of halophilic microorganisms and identification of their valuable biological agents using sequencing and genome mining tools.

From the title of this manuscript, I expected an in-depth review of the unique halophilic microorganisms that produce metabolites/products/enzymes that can be exploited in industries ranging from biotechnology to pharmaceuticals. However, there are only four paragraphs dedicated to this. Most of the manuscript attempts to provide background on halophiles, and to describe bioinformatics tools that are available. I do not believe the manuscript, as written, provides any unique interpretation of previous results, nor does it thoroughly compile the research on halophiles to date. Unfortunately, it does not offer anything of substance to the reader.

Too much emphasis is placed on the inability of some organisms to be cultured – this is a known fact, one sentence on this would suffice. Instead, lines 209-251 offer a repetitive discussion.

In addition, the paragraphs mentioning bioinformatics tools and methodologies and the brief discussion of how they work (Lines 279-294, section “Characteristics of nucleic acid sequencing methods and bioinformatics tools used in the biodiversity of saline environments”) is wholly unnecessary. I assume the authors do not intend for this manuscript to be a thorough review of bioinformatics toolsets, as it is not written as such, and thus these paragraphs are all general knowledge known in the field of bioinformatics. None of the information given is unique to halophiles and does not enhance the manuscript in any way. It would serve the authors to remove this large section entirely and focus more on what bioinformatics analyses have already proven true for halophilic species.

Unfortunately, the cited work by Singh and Singh (2017), mentioned as a review and titled “Haloarchaea: worth exploring for their biotechnological potential” serves as a better reference for the potentially exploitable properties that make halophiles worth studying than this manuscript currently does.

Much of the manuscript is repetitive and provides unnecessary information that either consists of widely-known aspects of culture-dependent and culture-independent analyses or provides unnecessary methodology when discussing specific examples of useful halophilic biomolecules.

There are only six examples with a very brief discussion of compounds of interest isolated/identified within halophiles. I would expect a much more thorough and in-depth analysis of the halophilic research conducted to date, what stands out, what is lacking, and what could improve research in the field.

Line items:

Line 33: It does not appear that Eukarya are discussed in the manuscript. Thus, while it is important to note that halophiles are found in all three domains of life, I would add a sentence that Eukaryotic halophiles will not be explored further in this manuscript.

Line 40-41: provide a citation for the definitions of slight, moderate, and extreme halophiles.

Line 42: Be more clear about what differentiates halotolerant species from halophilic species. From the sentences given, I see no difference.

Line 65: Please clarify the salinity of the oceans (i.e. 3.5% dissolved salts), as salinity is typically reported as 35 parts per thousand (ppt). Please also provide a reference for this statement.

Line 78: “GLS” is likely supposed to be GSL.

Line 78: Reference 15 is not about the GSL. If this is where the claim that GSL is the fifth saltiest water body in the world, the list in Table 1 in Ref. 15 does not take into account the ranges in salinity of most of the bodies listed.

Line 88: “athalassohaline” is not spelled correctly.

Line 102: Can you provide a comparison to the “5 x 105 cells/ml on average” found in the Dead Sea. How does this compare to other salty bodies? To Antarctica?

Lines 108, 109: “43,3%” and “40,3%” need to replace the commas with periods.

Line 147-148: This is already stated in the first sentence of the introduction.

Lines 173-175: what were the main conclusions of this research (ref. 83)?

Line 199: Provide a reference number for Singh and Singh 2017 so it can easily be found in the references list.

Lines 199-204: This does not appear to be true. Considering this manuscript has over 200 references, it is hard to believe that halophiles have been overlooked or unexplored. The references cited in refs. 109-113 seem to indicate otherwise.

Table 1: Why were these halophiles chosen? It would be helpful to include a column that explains whether they are bacteria or archaea, as well as a column that includes their optimal salt range. In addition, it is not clear why some of the listed attributes are given. For example, why is “iron Fe(II)” listed for Acidiphilium spp.? Do these organisms exists in high concentrations of iron?

Line 214: Why is reference 30 cited here when many others aren’t? I am sure that more than one of the references listed used culture-dependent approaches.

Line 216: Same question as line 214. Was reference 129 the ONLY paper that used cultivation of pure bacterial cultures?

Line 221: What is culturomics?

Line 224: It was just stated that culture-dependent methods don’t always work, why would you continue to list culture-based experimental approaches here?

Lines 226-233: These sentences are a slightly reworded replica of Lines 210-216.

Lines 239-245: These sentences are a slightly reworded, if at all, replica of lines 217-232.

Lines 255-256: proteomics and metabolomics are different than metagenomics and metatranscriptomics.

Line 257-259: this sentence does not describe functional metagenomics.

Line 282: Remove “sequencing” as it is already captured in NGS.

Line 288: “DNBSEQ-G400” by MGI appears to be a relatively new technology. Can you provide any references that use this technology for sequencing?

Lines 281-297: None of this background is necessary. Nucleic acid sequencing is a highly-utilized and well-developed technology.

Lines 298-331: This paragraph is also not necessary. This manuscript is not a review of the types of sequencing available and the information stated in this paragraph is fairly well-known.

Lines 332-394: Again, none of this is necessary. If the goal of this manuscript is to explore different bioinformatic strategies, a much more in-depth approach is necessary. Additionally, I would not necessarily say that Kraken2 and MetaPhlAn2 are the most frequently used tools. Is there any reference or statistics to back up this claim? Further, lines 359-365 are not necessary as well – this is standard general knowledge of bioinformatics. The focus of this manuscript does not appear to be on explaining bioinformatics tools and how they work.

Line 366: “analyzes” should be “analyses”. This appears multiple times throughout the manuscript.

Line 383: What is a lasso peptide?

Line 385: Explain what lantibiotics are, otherwise it looks like a typo.

Figure 1: I would not consider MGI a popular platform. Also, ThermoScientific is a science supply company, not a sequencing company. Should the first bullet point in the “long reads assembly” column read “short DNA fragments”? In the last bullet point of the marker gene sequencing column, what is ITS?

Line 397: what about all the other references that characterize halophilic metabolites?

Line 408: Why do we care about ectoine? What can we use it for?

Line 413: Why do we care about carotenoids? Have halophiles been used as microbial chassis for the production of these biomolecules for use in cosmetics or other industries?

Line 420-421: Like what? What are some antimicrobial or anticancer compounds and what have those researchers discovered about them?

Lines 422-426: This paragraph is repetitive and unnecessary.

Line 429: Citation needed.

Line 431-433: This sentence does not make sense. What is the first cation of the S. proteolyticus strain?

Line 442: what is the classic counterpart to PHA?

Line 454: Citation needed.

Line 455: Genes are typically written lowercase and italicized.

Lines 456-457: What were the results of the extensive physico-chemical analysis?

Line 463: Citation needed. There is no citation in this entire paragraph. Citation 139 is given at the end of the paragraph, but I believe this paragraph refers to Reference 138 (cited at line 480, but needs to be included further up as well).

Lines 464-473: These methods are unnecessary.

Line 481: citation needed.

Lines 483-488: These methods are unnecessary. The reader can access the original paper if they are interested in the specific methods. The purpose of this manuscript should be to highlight the previous research findings and explain why they are beneficial/of use.

Line 497: “In this field” does this refer to halophilic species? No antimicrobial peptides were discovered in halophilic species prior to 2015? I don’t believe this is the case.

Line 498: Citation needed.

Line 501-502: This sentence is not necessary.

Line 506: citation needed.

Lines 506-509: Why are these listed BGCs important?

Line 510: Citation needed.

Lines 512-517: Why is this important?

Line 525: “MAGs” should not have an apostrophe.

Line 535: Are Asgard archaea, specifically, Lokiarchaeota and Thorarchaeota, known halophiles? Provide a reference. Lines 533-542 appear to be unnecessary.

Lines 560-576: This is not a review on sequencing techniques or bioinformatics tools. One sentence would suffice to say that new methods/technologies would ultimately help in isolating and characterizing more halophiles.

Line 567-572: No where else in the manuscript is AI mentioned. It seems odd that it is included here.

Line 582-583: This sentence completely contradicts the one before it saying that halophiles have a “low level of exploration”. The 200+ references also contradict this statement.

References:

Ref. 58, 61, 195, 197 are not cited correctly.

Ref. 71, 198: no journal is given.

Round 2

Reviewer 1 Report

The revised version of the manuscript represents a major improvement. All my concerns were adressed.

Reviewer 2 Report

I appreciate the authors' responses to my initial comments and their attempts to address them. However, my main arguments against publication of this manuscript remain. 

As previously noted, from the title of the manuscript, as well as the final sentence of the abstract that proclaims that "the main aim of this review is to highlight the culture-independent experimental strategies used in halophile studies in concert with the presentation of recent examples of bioproducts and functions discovered in silico in the halophile's genomes", I expected an in-depth review of the unique halophilic microorganisms that produce metabolites/products/enzymes that can be exploited in industries ranging from biotechnology to pharmaceuticals. However, there are only about six examples provided. Most of the manuscript attempts to provide background on halophiles and halophilic environments, yet it is still not entirely thorough. Again, I would have expected the review to contain a much more in-depth analysis of the halophilic research conducted to date, what stands out, what is unique, what is lacking, and what could be done to improve research in the field.

I do not believe the manuscript, as written, provides any unique interpretation of previous results, nor does it thoroughly compile the research on halophiles to date. Unfortunately, it does not offer anything of substance to the reader that cannot be found elsewhere.
